# A Dynamics Coordinated Control System for 4WD-4WS Electric Vehicles

**Shaopeng Zhu [1], Bangxuan Wei [1], Dong Liu [2], Huipeng Chen [3], Xiaoyan Huang [1], Yingjie Zheng [3] and Wei Wei [4,*]**

[1] Power Machinery & Vehicular Engineering Institute, College of Energy Engineering, Zhejiang University, Hangzhou 310058, China
[2] Quanxing Machining Group Co., Ltd., Shaoxing 311800, China
[3] School of Mechanical Engineering, Hangzhou Dianzi University, Hangzhou 310018, China
[4] College of Information Engineering, Zhejiang University of Water Resources and Electric Power, Hangzhou 310018, China
[*] Correspondence: weiw@zjweu.edu.cn; Tel.: +86-138-1917-3537

**Abstract:** With the aggravation of the energy crisis and environmental problems, the new energy electric vehicle industry has ushered in vigorous development. However, with the continuous increase in car ownership, traffic accidents and other issues have gradually attracted widespread attention. Some existing stability coordination control systems often have problems, such as single stability judgment method and strong coupling between different subsystems. Therefore, based on previous research, it is necessary to further optimize the method of judging the vehicle's stability state, establish clear coordination rules, and reasonably solve the coupling problem between subsystems. This is of great significance for promoting the further development of the electric vehicle industry. Due to four-wheel-distributed driving and four-wheel-distributed steering electric vehicles having the characteristics of integrated driving, flexible steering, and easy fault-tolerant control, it has unique advantages in improving vehicle stability and is a good carrier for designing and constructing the stability coordination control system. In this paper, four-wheel-distributed driving and four-wheel-distributed steering (4WD-4WS) electric vehicles are taken as the research object, and a coordinated control strategy of four-wheel steering and four-wheel drive is proposed. Firstly, in order to realize the accurate judgment of vehicle stability, based on the vehicle two-degree-of-freedom two-track model and magic tire model, this paper uses the phase plane law to divide the phase plane stability region of the vehicle and introduces the stability quantification index PPS-region for the evaluation of vehicle stability. Secondly, a fuzzy variable parameter active rear-wheel steering controller and a compensated yaw moment controller are designed. Then, for the coupling problem between the two controllers, a coordination rule is proposed based on the stability index PPS-region of the phase plane stability region. Finally, a hardware-in-the-loop testbed is built to verify the feasibility of the coordination control strategy proposed in this paper. Experimental results show that: When the vehicle is in different stable states, according to the divided steady state, the control strategy can be correctly switched to the corresponding control strategy, and the work of each subsystem can be reasonably coordinated. Under the continuous gain sine condition, the control algorithm can reduce the maximum amplitude of the yaw rate error response curve by 73% and the side slip angle error response curve by 85%. Compared with a single stability control system, the coordinated stability control algorithm can improve the control effect of yaw rate and side slip angle by 20% and 62.5%. In the case of double lane-change, the control algorithm can reduce the maximum amplitude of the yaw rate error response curve by 68.5% and the side slip angle error response curve by 57.4%. Compared with a single stability control system, the coordinated stability control algorithm can improve the control effect of yaw rate and side slip angle by 40.6% and 44.7%.

**Keywords:** 4WD-4WS electric vehicles; coordinated control; phase plane analysis; active rear wheel steering control; direct yaw moment control

## 1. Introduction

With the continuous depletion of fossil natural energy, all countries face serious problems. Developing new energy electric vehicles is one of the effective ways to solve this problem. However, with the gradual increase in cars in recent years, related safety issues such as traffic accidents have attracted much attention. Therefore, the design and development of an advanced and efficient stability control system are of great significance for ensuring driving safety and promoting the development of the new energy vehicle industry.

Currently, a single chassis control system has gradually been challenging to meet the driver's pursuit of car performance. Giving full play to the advantages of various stability control subsystems and coordinating the functions of each system to improve vehicle handling stability and safety further has become one of the critical goals of the future development of automotive active safety technology. Four-wheel-distributed driving and four-wheel-distributed steering electric vehicles use drive motors and steering motors as actuators. This type of vehicle has the characteristics of integrated driving and braking, flexible steering, and easy failure-tolerant control. It has significant advantages in improving vehicle stability and is a suitable carrier for designing and constructing stability-coordinated control systems [1,2]. Therefore, making full use of this type of electric vehicle's advantages in driving and steering, a new design scheme of stability coordination control system is proposed and verified by experiments. This has profound significance for promoting industrial development and providing new ideas for technological innovation.

According to different control methods, the vehicle stability control system is divided into the longitudinal force control system, vertical force control system, and lateral force control system. Since this study focuses on the longitudinal force control system and the lateral force control system of the automobile, it will analyze and discuss the longitudinal force control and the lateral force control. Consequently, the vertical force control system of the vehicle will not be discussed in detail.

In terms of driving torque distribution, for the stability control of distributed-driven electric vehicles, researchers often choose the combination of the yaw rate and the side-slip angle or one of them as the control target variable. Hence, based on the total driving force required for vehicle driving and the yaw moment needed to maintain stability, Reference [3] designed a torque average distribution strategy that can be quickly calculated and responded to in time. Moreover, Reference [4] proposed a hierarchical control strategy for the electric vehicle driving force that considers the wheel load difference between the front and rear axles. Based on the fuzzy control algorithm, the control strategy formulated the compensating yaw moment required by the vehicle and adjusted the torque according to the load difference of each wheel. Reference [5] proposed an optimal torque distribution strategy based on the tire friction circle theory. This strategy took the stability margin of the tire as the objective optimization function and solved the objective function through optimization algorithms such as quadratic programming.

In terms of steering, the traditional four-wheel steering (4WS) vehicle stability control system mainly refers to the active rear-wheel steering system (ARS). For the design of the active rear-wheel steering system (ARS), domestic and foreign researchers have carried out extensive research. Among them, Reference [6] suggested a control strategy to control the steering angle of the rear wheel by changing the pulse signal. After qualitative testing and analysis of the pulse parameters, those showing the best performance were selected after comparison. After verification via CarSim and Matlab co-simulation, an SUV vehicle was modified; the effectiveness of the controller was verified through vehicle testing and analysis, and the yaw stability performance of the vehicle was enhanced. Reference [7] designed the $\mu$ controller and the $H_\infty$ controller using the model matching method, which improved the handling stability of the vehicle. Furthermore, Reference [8] proposed an active rear-wheel steering control strategy based on a fuzzy inference system. For the understeer state and oversteer state of the vehicle, the relevant fuzzy control sets were set according to the driving speed. Afterwards, the impact of the ARS system on vehicle stability was analyzed from the perspectives of time and frequency domains.

In the aspect of the integration of stability control system, there are also some good research results. Reference [9] suggested a vehicle stability control strategy that combines rear-wheel active steering and differential braking to generate a compensating yaw moment and designed a fuzzy controller for the vehicle stability control system with multi-variable input and output. A genetic algorithm optimized the fuzzy controller parameters and membership function distribution, and the transient response performance of the system was improved. Reference [10] used ABS and DYC as control methods, designed a dual-motor distributed drive system collaborative control method, and applied it to FSAE racing cars. Reference [11] proposed a vehicle stability control strategy based on the synergistic effect of active rear-wheel steering and single-wheel differential braking. Firstly, the optimal following controller for active steering of rear wheels and the fuzzy controller for vehicle stability was designed. Then, a coordination scheme was proposed to distinguish the respective control tasks in the form of a fuzzy control membership function. The effectiveness of the control strategy was best verified by using software simulation. The above literatures have integrated designs for different stability systems. From the experimental results in the literature, it can be found that, compared with a single stability system, the stability integrated system can improve the stability and working condition adaptability of the vehicle to a certain extent. These research results provide a certain theoretical basis for the smooth development of this research.

At present, many experts and scholars have also achieved many excellent research results in the design of the stability system of four-wheel-distributed driving and four-wheel-distributed steering (4WD-4WS) electric vehicles. The technical scheme is shown in Table 1. Based on fuzzy control theory, Reference [12] uses the side slip angle and yaw rate as control variables to control the rear wheel rotation angle and additional yaw moment required for the vehicle to steer stably. The additional yaw moment is distributed to the four driving wheels using the differential driving moment of the left and right wheels.

**Table 1.** Technical solutions.

| Control Objectives | Control Method | References |
|---|---|---|
| side slip angle, yaw rate | ARS + DYC | [12] |
| side slip angle, yaw rate | ARS + DYC | [13] |
| side slip angle, yaw rate | ARS + DYC | [14] |
| side slip angle, yaw rate | AFS + DYC | [15] |
| side slip angle, yaw rate, vehicle speed | ARS + DYC | [16] |
| side slip angle, yaw rate | AFS + DYC | [17] |

Taking advantage of the fact that the ARS can reduce the vehicle's side-slip angle and the DYC is able to design the vehicle's steady-state steering characteristics, Reference [13] designed a fuzzy PID controller, coordinated by ARS and DYC. The purpose is to make the vehicle's yaw rate follow the ideal value while reducing the vehicle's side slip angle. On the other hand, Reference [14] constructed a rear-wheel steering controller and a vehicle stability controller based on a linear vehicle model and suggested a coordinated control strategy for active rear-wheel steering and torque distribution to improve vehicle handling stability. Reference [15] adopts an improved sliding mode control to design an integrated control strategy for AFS and DYC. Proportional-integral (PI) sliding mode surface is introduced, and the side slip angle and yaw rate are used as control targets. The experimental verification shows that the vehicle has good robustness in the presence of external disturbances and uncertain model parameters. Furthermore, Reference [16] first designed a feedforward + feedback active rear-wheel steering controller with the

vehicle's side-slip angle as the control target. Subsequently, a four-wheel torque distribution controller was constructed with the vehicle yaw rate and desired longitudinal vehicle speed as the control targets. Finally, a rule-based coordination controller was designed to allocate the operating range of each sub-controller, reasonably. Reference [17] designs a front wheel active steering controller based on sliding mode control and a direct yaw moment controller based on the model predictive control method. In order to coordinate the work of two different controllers, the phase plane method is used as the judgment basis to allocate the weights of each controller adaptively to realize the switching between controllers.

The above literature provides a rich theoretical basis and design ideas for the design of the stability coordination system of the four-wheel-distributed driving and four-wheel-distributed steering (4WD-4WS) electric vehicles and has also achieved many excellent results. However, in these studies, there is still strong coupling between different systems, and in some cases, the advantages of individual subsystems may not be fully exploited. Moreover, the research on the coordination principles and strategies among the various subsystems is not deep enough. For example, most literature uses the thresholds of vehicle speed and stability state parameters as the judgment basis. Because the selection of thresholds is usually highly subjective, the theoretical basis is not strong. In addition, due to the complex and changeable driving environment of the vehicle, the adaptability of the vehicle under multiple operating conditions needs to be improved. Therefore, based on the above literature, these studies still have the following room for optimization and improvement.

Based on the above literature, researchers have conducted extensive studies on drive torque distribution, active rear-wheel steering control, and coordinated control of the aforementioned two different control systems. However, these studies still have the following room for optimization and improvement. First, regarding the judgment of vehicle stability, the method adopted by most literature is to use the difference between the yaw rate/center slip angle and their respective expected values to characterize the vehicle stability. Nevertheless, relying exclusively on the values of the two variables, the center of mass slip angle and the yaw rate can only be used to preliminarily judge the stability of the vehicle. To further and accurately judge the stability of the vehicle, the follow-up study can use the phase plane method to divide the vehicle stability region. In addition, based on the stability parameters, a computable quantitative factor can be introduced to make a specific judgment on the vehicle stability state. Second, regarding the active rear-wheel steering control, the above literature proposed different control strategies and optimization methods to improve the operational stability of the vehicle. In terms of enhancing the robustness of the controller, multiple researchers have also conducted related studies. Nonetheless, the previous study mainly focused on the perturbation of model parameters caused by external disturbance factors (such as lateral wind). However, changes in road adhesion conditions will also affect the vehicle dynamics, which may further limit the effect of the rear-wheel active steering system in improving vehicle stability to some extent. Finally, regarding coordinated control systems, much of the aforementioned literature has room for further optimization and improvement in coordination principles/strategies. For example, some literature did not propose explicit coordination rules but simply superimposed the drive torque distribution system and the active rear-wheel steering control system. Subsequently, this kind of control system will have a strong coupling phenomenon in various working conditions and will not be able to give full play to the advantages of each subsystem. Other literature only used the threshold value of vehicle speed and yaw rate as the direct basis for coordination system decision-making. However, the threshold selection is usually highly subjective, and the theoretical basis was not solid. The majority were obtained through simulation test analysis or experience, which also indirectly restricted the performance of the integrated control system.

To further ensure the driving safety of drivers and passengers at high speed and give full play to the advantages of this type of electric vehicle in stability control, this study suggests a coordinated control strategy of four-wheel steering and four-wheel drive. Firstly, based on the phase plane division rule, the phase plane of "side slip angle-side slips

angular velocity" was drawn. Moreover, the phase plane stability domain was divided according to the double straight-line method. In agreement with the division results, the boundary model was designed, and the boundary coefficients were introduced. On the basis of analyzing the influence of vehicle speed and road adhesion coefficient on the boundary of the stability domain of the vehicle phase plane, and according to the simulation results, a mapping data table between the boundary coefficient of the stability domain and the two influencing factors was established. Afterwards, the boundary of the stability domain of the vehicle system was calculated. To accurately characterize the stability state of the vehicle, based on the boundary of the stability domain, the phase plane was further divided into three parts: the stable region, the critical region, and the unstable region; the phase plane stability index (PPS-region) was introduced to characterize the vehicle stability state quantitatively. Second, a dynamic coordinated control strategy was implemented. The strategies included a VLQR-based active rear-wheel steering control, an FC-based compensation yaw moment control, a coordinated control strategy, and a drive/brake torque distribution module. Based on the difference between the ideal value and the actual value of the yaw rate and the side-slip angle of the center of mass, an LQR controller was established. Then, combined with the road adhesion coefficient, the fuzzy control variable weight coefficient regulator was added to form the active rear-wheel steering controller of the VLQR. This controller is able to dynamically adjust the importance of the LQR controller to the side-slip angle and the yaw angular velocity through the fuzzy control variable weight coefficient regulator, to increase the controller's adaptability to different road conditions. Afterwards, the FC-based compensating yaw moment controller was set up, and the desired compensation yaw moment of the vehicle was acquired by combining the deviation of the actual and ideal values of the yaw rate and the sideslip angle. Subsequently, an appropriate control scheme was given, based on the coordinated control rules and the real-time vehicle state. After calculation, the additional yaw moment of driving/braking and the rotation angle of the rear wheels could be obtained. Through the driving torque distribution module or the braking torque distribution module, the additional yaw moment could be combined with the total driving force/total braking force to acquire the driving torque or braking torque of each wheel, and eventually control the vehicle to adapt to different stability domain states and driving conditions.

Finally, through CAN communication and the DSP controller, the hardware-in-the-loop test platform was built using an NI PXIe-8880 real-time controller and NI PXI-8512 board to complete the feasibility verification of the coordination control strategy proposed in this study. Experimental results show that: When the vehicle is in different stable states, according to the divided steady state, the control strategy can be correctly switched to the corresponding control strategy, and the work of each subsystem can be reasonably coordinated. Under the continuous gain sine condition, the control algorithm can reduce the maximum amplitude of the yaw rate error response curve by 73% and the side slip angle error response curve by 85%. Compared with a single stability control system, the coordinated stability control algorithm can improve the control effect of yaw rate and side slip angle by 20% and 62.5%. In the case of double lane-change, the control algorithm can reduce the maximum amplitude of the yaw rate error response curve by 68.5% and the side slip angle error response curve by 57.4%. Compared with a single stability control system, the coordinated stability control algorithm can improve the control effect of yaw rate and side slip angle by 40.6% and 44.7%.

A list of abbreviations is shown in Table 2.

**Table 2.** Abbreviation table.

| Abbreviation | Meaning |
| --- | --- |
| 4WD-4WS | Four-wheel driving and Four-wheel steering |
| PPS-region | An indicator of stability |
| 4WS | Four-wheel steering |
| ARS | Active rear-wheel steering system |

**Table 2.** *Cont.*

| Abbreviation | Meaning |
|---|---|
| DYC | Direct yaw moment control |
| ABS | Anti-lock braking system |
| AFS | Active Front Wheel Steering |
| VLQR | Varying parameter Linear Quadratic Regulator |
| LQR | Linear Quadratic Regulator |
| FC | Fuzzy control |
| FWS | Front wheel steering |

## 2. Segmentation of Stability Domain of the Side-Slip Angle Based on Phase Plane Law

### 2.1. Establishment of the Phase Plane Diagram

The difference between the ideal and actual values of the yaw rate, as well as the difference between the ideal and actual values of the side-slip angle, is often used to characterize the vehicle's stability. However, these two values can only be used for the preliminary judgment of vehicle stability. To further accurately determine the stability state of the vehicle, based on the differential equations of vehicle dynamics, this study used the phase plane law to delineate the phase plane stability region of the vehicle to accurately determine the aforementioned stability [18].

A stability region exists in the phase plane diagram, in which the phase trajectories from any initial point eventually converge to a stable focal point, and then the vehicle can recover to a stable state. To adapt to the complex and changing vehicle driving conditions, this study selected the phase plane of "side slip angle-side slips angular velocity" to study vehicle stability, which is less affected by the vehicle speed. Subsequently, the change of the stability domain boundary is easy to fit under different road surface adhesion conditions and the front-wheel steering angle [19–21].

Based on the phase plane analysis method, a two-degree-of-freedom two-track model of the vehicle is established, as shown in Figure 1.

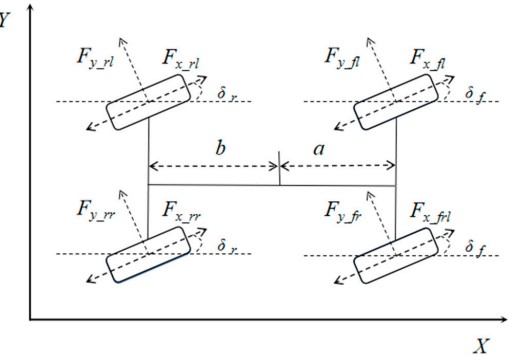

**Figure 1.** Vehicle dynamics model.

Its dynamic differential equation is expressed as [22,23]:

$$\begin{cases} \sum F_Y = cos\delta_f \left( F_{y\_fl} + F_{y\_fr} \right) + cos\delta_r \left( F_{y\_rl} + F_{y\_rr} \right) \\ \sum M_z = F_{y\_fl} \left( acos\delta_f + \frac{w}{2}sin\delta_f \right) + F_{y\_fr} \left( acos\delta_f - \frac{w}{2}sin\delta_f \right) + \\ \qquad F_{y\_rl} \left( bcos\delta_r - \frac{w}{2}sin\delta_r \right) - F_{y\_rr} \left( bcos\delta_r + \frac{w}{2}sin\delta_r \right) \end{cases} \tag{1}$$

where: $F_{y\_fl}$—The lateral force provided by the left front wheel; $F_{y\_fr}$—The lateral force provided by the right front wheel; $F_{y\_rl}$—The lateral force provided by the left rear wheel; $F_{y\_rr}$—The lateral force provided by the right rear wheel; $w$—Wheel distance; $a, b$—The distance from the center of mass to the front axle and the distance from the center of mass

to the rear axle; $\delta_f$, $\delta_r$—The steering angle of the front wheel and the steering angle of the rear wheel.

Since the longitudinal vehicle speed is assumed to be constant when analyzing the phase plane of "Side slip angle-Side slips Angular velocity", the longitudinal tire force is not considered here, and only the tire lateral force is taken into account.

The magic tire model has excellent simulation performance of the whole vehicle operation and stability, so this paper uses the simplified magic formula to express the simplified lateral force of the tire as [24]:

$$F_{y\_ij} = \mu F_Z sin(D \, arctan(B\alpha_{ij})) \tag{2}$$

where: $F_{y\_ij}$—Lateral force of each tire; $\mu$—the road adhesion coefficient; $F_{Z\_ij}$—Vertical load of each tire; $B$, $D$—Coefficient to be fitted; $\alpha_{ij}$—Side slip angle of each tire.

The formula for calculating the side slip angle of each wheel is

$$\begin{cases} \alpha_{fl} = arctan(\frac{V \, sin\beta + a\omega}{V \, cos\beta - 0.5 \, w\omega}) - \delta_f \\ \alpha_{fl} = arctan(\frac{V \, sin\beta + a\omega}{V \, cos\beta - 0.5 \, w\omega}) - \delta_f \\ \alpha_{rl} = arctan(\frac{V \, sin\beta - b\omega}{V \, cos\beta - 0.5 \, w\omega}) - \delta_r \\ \alpha_{rr} = arctan(\frac{V \, sin\beta - b\omega}{V \, cos\beta + 0.5 \, w\omega}) - \delta_r \end{cases} \tag{3}$$

where: $V = \sqrt{V_x^2 + V_y^2}$—Speed of the vehicle; $\beta$—Side slip angle; $\omega$—Yaw rate.

The differential equation is obtained by associating the vehicle two-degree-of-freedom two-track model with the magic tire model under the condition of constant vehicle speed.

$$\begin{cases} \dot{\beta} = \frac{\sum F_Y}{mV - \omega} \\ \dot{\omega} = \frac{\sum M_Z}{I_z} \end{cases} \tag{4}$$

where: $I_Z$—Moment of inertia of the vehicle around the $Z$ axis;

Based on the above analysis, the phase plane model of the side slip angle of mass is constructed in MATLAB/Simulink. The phase trajectory motion curves of "Side slip angle-Side slips Angular velocity" are obtained by the initial state of the given vehicle dynamics model.

### 2.2. Division of Stability Domains in Phase Plane

In this paper, the bilinear method is used to partition the stability domain of the phase plane, as shown in Figure 2. This method usually represents the boundary lines of the stability domain as two straight lines symmetric about the origin. These two lines pass through the saddle point and are tangent to the convergent critical trajectory.

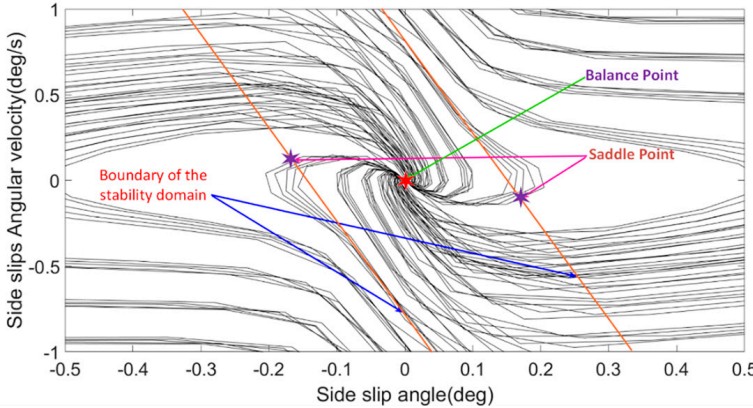

**Figure 2.** Schematic diagram of the stable domain of the phase plane divided by the bilinear method.

The mathematical model of its stable domain region can be expressed as [25–27]:

$$B_1 \leq \dot{\beta} + A\beta \leq B_2 \qquad (5)$$

where: $A, B_1, B_2$—Boundary coefficient; A denotes the slope of the stable domain boundary; $B_1$ is the slope of the upper boundary of the stable domain, and $B_2$ is the slope of the lower boundary of the stable domain.

Based on the formula in Section 2.1, the phase plane model of the side slip angle is established in MATLAB/Simulink. By assigning different values to the initial states of the state equation, the phase locus diagrams under different working conditions can be obtained. Next, this paper will analyze the influence of road adhesion coefficient and vehicle speed on the boundary coefficient:

(1) When the vehicle speed is constant, the influence of the road adhesion coefficient on the boundary coefficient:

Under the condition that the vehicle speed is constant at 100 km/h and the turning angle of the front wheels is zero, the simulation experiment is carried out in the interval of the road adhesion coefficient [0.1, 1]. The phase diagram obtained by the simulation is divided by the double straight-line method, and the boundary parameters under different road adhesion coefficients can be obtained, as shown in Table 3.

**Table 3.** Boundary parameters under different road adhesion coefficients.

| Number | Road Adhesion Coefficient $\mu$ | $A$ | $B_1$ | $B_2$ |
|--------|--------------------------------|------|-------|-------|
| 1 | 0.1 | 2.32 | −0.13 | 0.13 |
| 2 | 0.2 | 3.86 | −0.22 | 0.22 |
| 3 | 0.3 | 4.04 | −0.36 | 0.36 |
| 4 | 0.4 | 4.53 | −0.50 | 0.50 |
| 5 | 0.5 | 4.94 | −0.56 | 0.56 |
| 6 | 0.6 | 5.17 | −0.63 | 0.63 |
| 7 | 0.7 | 5.47 | −0.70 | 0.70 |
| 8 | 0.8 | 5.96 | −0.81 | 0.81 |
| 9 | 0.9 | 6.13 | −0.97 | 0.97 |
| 10 | 1.0 | 6.54 | −1.08 | 1.08 |

It can be seen from Table 1 that at the same vehicle speed, with the decrease of the road adhesion coefficient, the range of the phase plane stable region gradually decreases.

(2) When the road adhesion coefficient is constant, the influence of vehicle speed on the boundary coefficient:

When the current wheel angle is 0, and the road adhesion coefficient is 0.8, in the interval of [50 km/h, 120 km/h], the vehicle speed is taken at intervals of 10 km/h, and the simulation is carried out. The phase diagram obtained by the simulation is divided by the double straight-line method, and the boundary parameters under different vehicle speeds can be obtained, as shown in Table 4.

**Table 4.** Boundary parameters at different speeds.

| Number | Vehicle Speed | $A$ | $B_1$ | $B_2$ |
|--------|---------------|------|-------|-------|
| 1 | 50 km/h | 6.11 | −0.83 | 0.83 |
| 2 | 60 km/h | 6.07 | −0.83 | 0.83 |
| 3 | 70 km/h | 6.08 | −0.81 | 0.81 |
| 4 | 80 km/h | 6.06 | −0.82 | 0.82 |
| 5 | 90 km/h | 6.02 | −0.83 | 0.83 |
| 6 | 100 km/h | 5.96 | −0.81 | 0.81 |
| 7 | 110 km/h | 6.01 | −0.84 | 0.84 |
| 8 | 120 km/h | 6.05 | −0.83 | 0.83 |

It can be seen from Table 4 that under the same adhesion coefficient, with the increase of the vehicle speed coefficient, the boundary curve of the stable region of the phase plane hardly changes. This simulation result is also consistent with the previous research conclusions of scholars.

Finally, after fitting the stability region boundary curve in the phase plane of "Side slip angle-Side slips Angular velocity" by the bilinear method, the expression of the boundary line of the stability region is obtained.

$$\begin{cases} B_2 \le \dot{\beta} + A\beta \le B_1 \\ A = -2.765\,\mu^2 + 7.073\,\mu + 2.07 \\ B_1 = -B_2 = 0.04167\,\mu^2 + 0.9675\,\mu + 0.04783 \end{cases} \tag{6}$$

### 2.3. Calculation of Phase Plane Stability Index (PPS-Region)

When the state point is near the boundary of the stability region, its convergence is slow. Moreover, during this slow convergence process, the vehicle may have been in danger. Therefore, this paper further divides the stable region into the stable region and critical region, as shown in Figure 3.

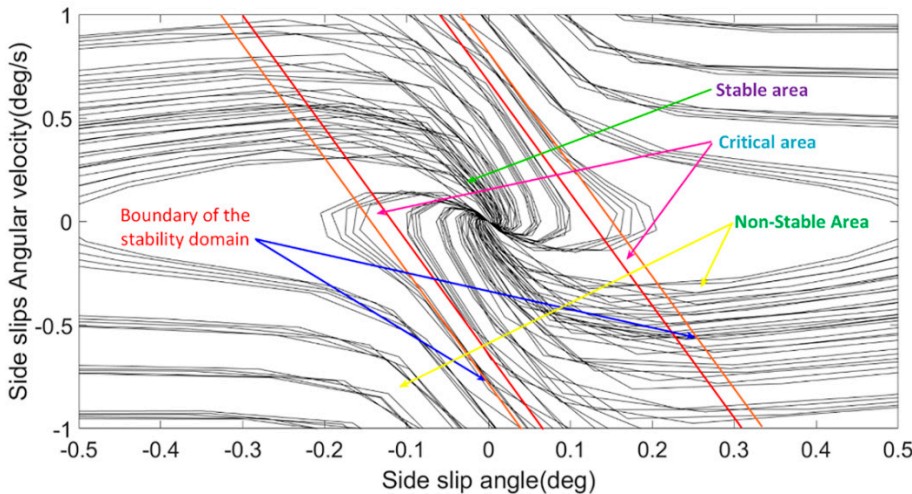

**Figure 3.** Further delineation of phase plane stability domain.

In order to facilitate the quantitative representation of the stability state of the state point in the region, this paper defines the phase plane stability (PPS_region) to represent the stability state in which the vehicle is located, and its expression is

$$PPS\_region = \left| \frac{1}{A}\dot{\beta} + \frac{B_1}{A}\beta \right| \tag{7}$$

The correspondence between the "PPS-region" and the steady-state the vehicle is in is

$$\text{Steady state of the vehicle} = \begin{cases} Non\text{-}Stable\ region, & PPS\_region > 0.8 \\ Critical\ region, & 0.8 \le PPS\_region \le 1 \\ Stable\ region, & PPS\_region < 0.8 \end{cases} \tag{8}$$

## 3. Dynamics Coordinated Control System

As shown in Figure 4, the dynamics coordination control system mainly includes the perception and judgment layer and the decision and control layer. In this setting, the main task of the perception and judgment layer is to recognize the driver's driving intention and calculate the vehicle's current phase plane stability "PPS-region", desired yaw rate, desired side slip angle, and desired total driving force based on external input and

feedback from the vehicle's current motion state. As for the control decision layer, it mainly includes an active rear-wheel steering control based on the VLQR (Varying parameter Linear Quadratic Regulator), a compensated yaw moment control based on FC (Fuzzy Control), and a dynamics coordination control based on the PPS-region. Depending on the phase plane stability PPS-region, the dynamics coordination controller performs a coordinated control distribution of the rear-wheel steering angle, four-wheel-drive torque, and four-wheel brake torque.

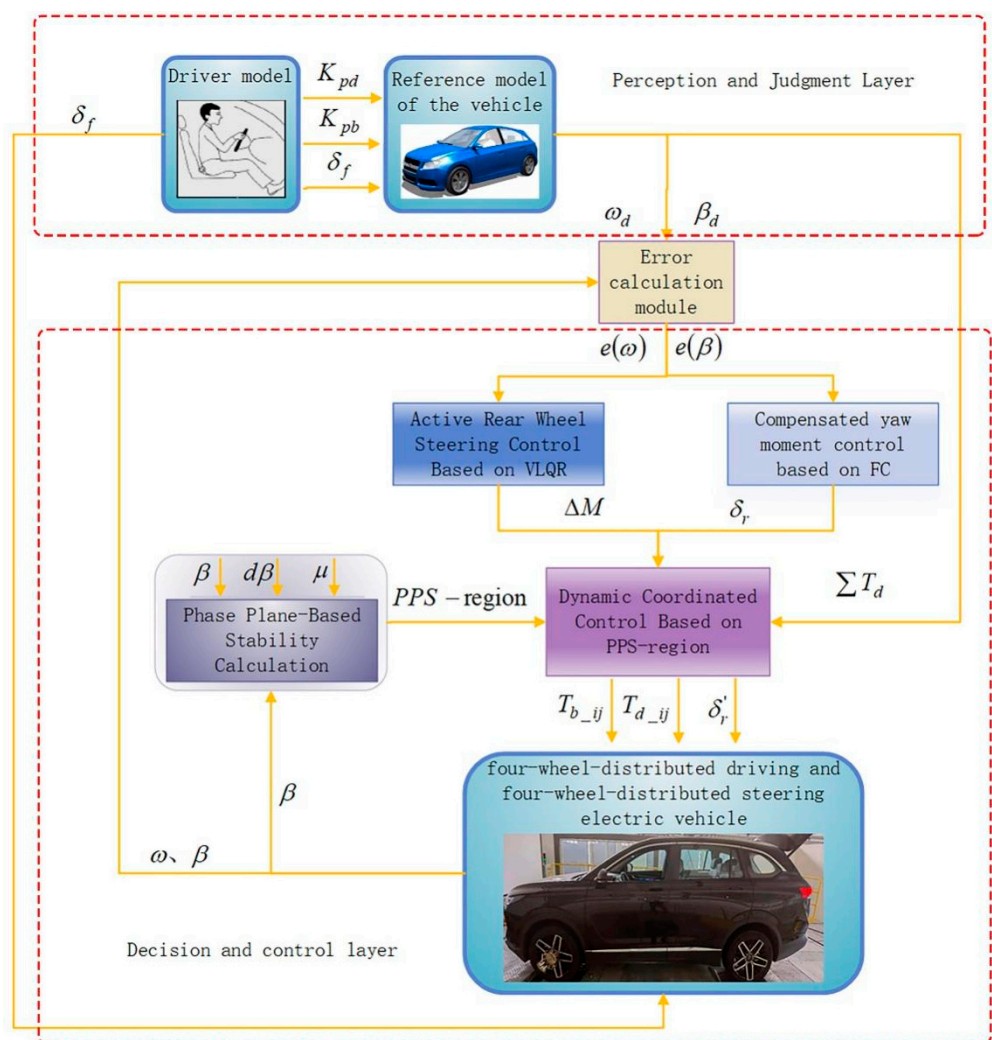

**Figure 4.** Structure block diagram of dynamics coordinated control system.

### 3.1. VLQR-Based Active Rear-Wheel Steering Control

In order to enable the vehicle to fully utilize the handling stability under different road conditions, a fuzzy variable parameter active rear-wheel steering controller (VLQR active rear-wheel steering controller) is designed in this paper, as shown in Figure 5.

First, input the difference value $e(\beta)$ between the ideal value and the actual value of the yaw rate and the difference value $e(\omega)$ between the ideal value and the actual value of the side slip angle into the LQR active rear-wheel steering controller. Then, the fuzzy control-based variable weight coefficient regulator will output different weight coefficients $q_\beta$ and $q_\omega$ to the LQR active rear-wheel steering module according to the variation of the road surface adhesion coefficient $\mu$. Finally, the LQR-based active rear-wheel steering controller calculates the optimal rear-wheel steering angle for the current vehicle driving conditions and adjusts the rear-wheel steering angle in real-time to achieve fully closed-loop control.

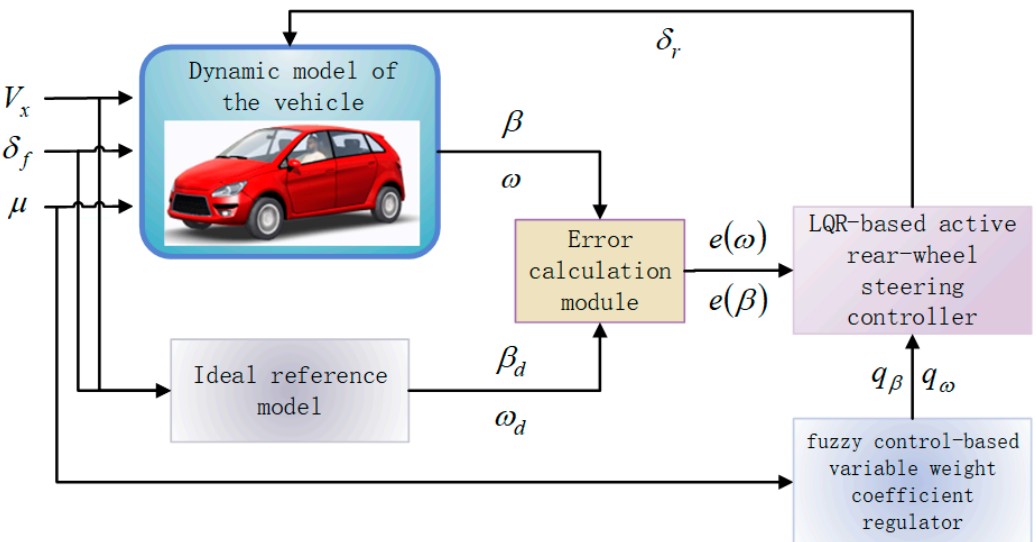

**Figure 5.** VLQR active rear-wheel steering controller.

According to the experience of previous scholars [28,29], the vehicle reference model takes the ideal yaw rate $\omega_d$, and the ideal side slip angle $\beta_d$ as the state variables. The deviation between the actual and ideal values of the state variables is

$$e = X - X_d = \begin{bmatrix} e_\beta \\ e_\omega \end{bmatrix} = \begin{bmatrix} \beta - \beta_d \\ \omega - \omega_d \end{bmatrix} = \begin{bmatrix} \beta \\ \omega - \frac{v_x}{(a+b)(1+Kv_x^2)} \end{bmatrix} \tag{9}$$

where: $V_x$—Longitudinal speed of the vehicle; $K$–The stability factor.

According to the linear quadratic regulator control theory, the control objective of the LQR controller is to find the optimal control output. That is, the actual output of the control system is as close to the expected output as possible within a certain time interval [30]. Let the state space expression of a linear time-varying system be [31]:

$$\dot{X} = AX + BU + CW \tag{10}$$

where:

$$A = \begin{bmatrix} -\frac{k_1+k_2}{mV_x} & -\frac{ak_1-bk_2}{mV_x^2} - 1 \\ -\frac{ak_1-bk_2}{I_z} & -\frac{a^2k_1+b^2k_2}{I_z V_x} \end{bmatrix} \tag{11}$$

$$B = \begin{bmatrix} \frac{k_2}{mV_x} \\ -\frac{bk_2}{I_z} \end{bmatrix}, \quad C = \begin{bmatrix} \frac{k_1}{mV_x} \\ \frac{ak_1}{I_z} \end{bmatrix} \tag{12}$$

$$\dot{X} = \begin{bmatrix} \dot{\beta} \\ \dot{\omega} \end{bmatrix}, \quad X = \begin{bmatrix} \beta \\ \omega \end{bmatrix} \tag{13}$$

$$U = [\delta_r], \quad W = \begin{bmatrix} \delta_f \end{bmatrix} \tag{14}$$

In order to find the optimal solution, the performance index $J$ of the LQR active rear-wheel steering controller is set [31], and the expression of $J$ is obtained by collating Equations (9) to (10) as

$$\begin{cases} J = \frac{1}{2} \int_0^\infty [(X - X_d)^T Q(X - X_d) + U^T RU] dt \\ Q = \begin{bmatrix} q_\beta & 0 \\ 0 & q_\omega \end{bmatrix}, R = [r] \end{cases} \tag{15}$$

where: $q_\beta$—weighting coefficient of the side slip angle, characterizing how much importance the controller places on the error value $e(\beta)$ of the side slip angle in the state variable;

$q_\omega$—weighting coefficient of the yaw rate, characterizing how much importance the controller places on the error value $e(\omega)$ of the yaw rate in the state variable; $r$—characterizes the extent to which the system limits the rear wheel rotation angle.

To find the optimal solution for LQR, the Hamiltonian function is constructed, and the Riccati equation is set as

$$Q + A^T + PA - PBR^{-1}B^T P = 0 \tag{16}$$

where $P$ is the solution of the Riccati equation.

After arranging Equations (10)–(16), the optimal control of the system can be obtained as

$$U^*(t) = -R^{-1}B^T PX + R^{-1}B^T(PBR^{-1}B^T - A^T)^{-1}(QA_d - PC)\delta_f \tag{17}$$

The feedforward gain matrix of the control system is

$$K_{FB} = -R^{-1}B^T P \tag{18}$$

The state feedback gain matrix of the control system is

$$K_{FF} = R^{-1}B^T(PBR^{-1}B^T - A^T)^{-1}(QA_d - PC) \tag{19}$$

It is not difficult to find through the construction process of the control system that the weighting matrices $Q$ and $R$ determine the performance and control effect of the linear quadratic regulator to a certain extent. However, selecting these two matrix parameters is usually based on the researcher's expertise. Hence, after selection, the controller's performance is evaluated and adjusted through design experiments. Note that such a selection process consumes a lot of resources, and the selected parameters cannot guarantee that the controller is capable of achieving the optimal control effect.

Therefore, when selecting the parameters of the weighting matrix, it should be adjusted according to the different driving conditions of the vehicle, to adjust the degree of importance that the controller attaches to the side slip angle and the yaw rate, to improve the efficiency of the parameter selection and the performance of the controller.

When the adhesion coefficient of the road surface is low, the vehicle stability remains more sensitive to the side slip angle [32–35]. On the other hand, when the road adhesion coefficient is high, the vehicle stability control is more concerned with the yaw rate. To make the LQR-based active rear-wheel steering controller maintain optimal performance under different road conditions, the following adjustment rules are established in this study.

Combining Table 5, the fuzzy inference rules are designed, and the fuzzy controller shown in Figure 6 is built. The working principle follows: Firstly, the road adhesion coefficient $\mu$ is input into the controller and fuzzed. After that, the fuzzy rules within the fuzzy controller are applied to perform fuzzy inference in combination with the input fuzzy variables. Finally, the results obtained by fuzzy inference are defuzzified using the area center of gravity method to obtain two output variables, the weight coefficient $q_\beta$ of the side slip angle and the weight coefficient $q_\omega$ of the yaw rate.

**Table 5.** Dynamic adjustment rules.

| Road Surface Adhesion Conditions | Take Value | Adjustment of Objectives |
|:---:|:---:|:---:|
| $\mu\downarrow$ | $q_\beta \uparrow$ <br> $q_\omega \downarrow$ | Control the side slip angle within a reasonable range |
| $\mu\uparrow$ | $q_\omega \uparrow$ <br> $q_\beta \downarrow$ | Reducing the error between the yaw rate and its ideal value |

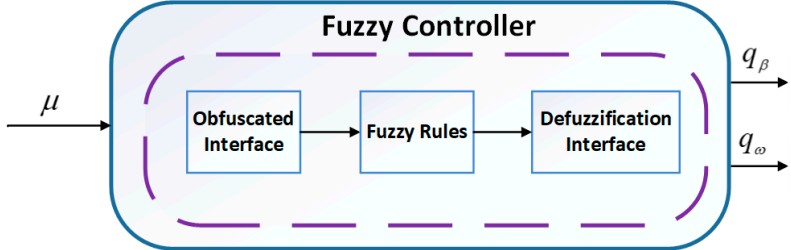

**Figure 6.** The fuzzy control-based variable weight coefficient regulator.

The input and output fuzzy sets are shown in Table 6. The membership function of $\mu$ is shown in Figure 7. The membership function of $q_\beta$ is shown in Figure 8. The membership function of $q_\omega$ is shown in Figure 9.

**Table 6.** Fuzzy set of input and output.

| $\mu$ | $q_\beta$ | $q_\omega$ |
| --- | --- | --- |
| NB | NB | NB |
| NM | NS | NS |
| NS | ZE | ZE |
| ZE | PS | PS |
| PS | PB | PB |
| PM | | |
| PB | | |

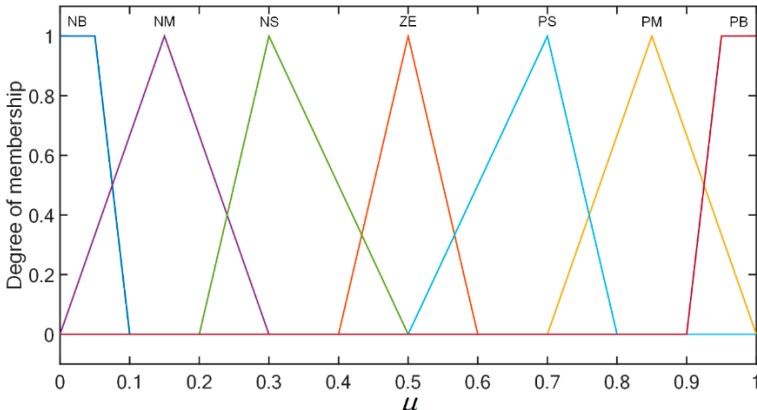

**Figure 7.** Membership function of $\mu$.

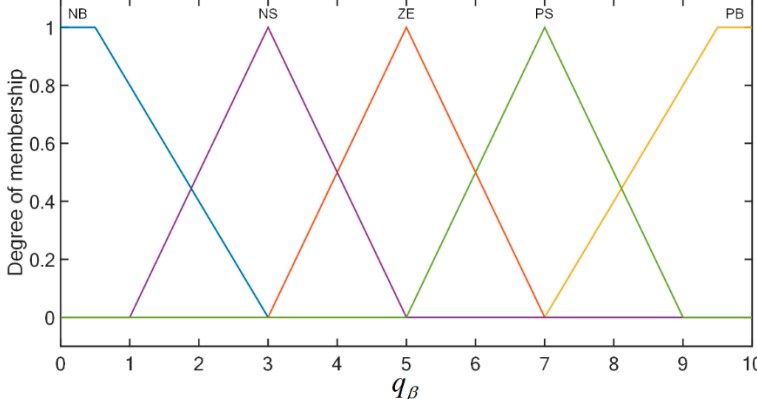

**Figure 8.** Membership function of $q_\beta$.

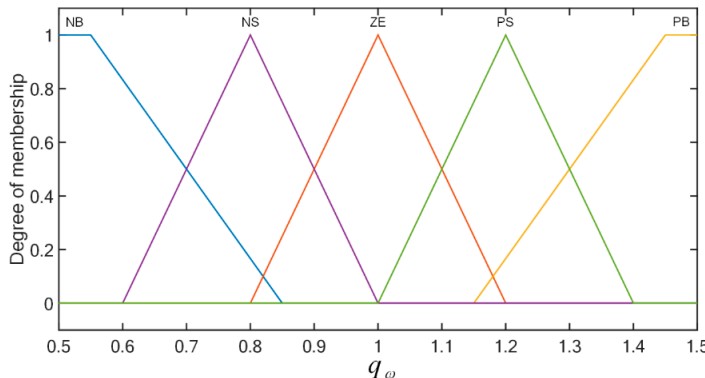

**Figure 9.** Membership function of $q_\omega$.

The fuzzy inference rule set is shown in Table 7.

**Table 7.** Fuzzy Inference Rule.

| $\mu$ | $q_\beta$ | $q_\omega$ |
|---|---|---|
| NB | PB | NB |
| NM | PS | NS |
| NS | PS | NS |
| ZE | ZE | ZE |
| PS | NS | PS |
| PM | NS | PS |
| PB | NB | PB |

*3.2. FC-Based Compensated Yaw Moment Control*

This paper designs a fuzzy (FC) controller, as shown in Figure 10. The controller takes the error quantity $e(\beta)$ of the side slip angle and the error quantity $e(\omega)$ of the yaw rate as the input quantities and the compensating yaw torque $\Delta M_w$ as the output variable.

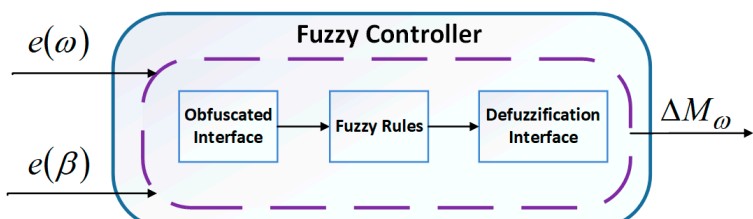

**Figure 10.** Compensated yaw moment controller.

The input and output fuzzy sets are shown in Table 8. The membership function of $e(\beta)$ is shown in Figure 11. The membership function of $e(\omega)$ is shown in Figure 12. The membership function of $\Delta M_w$ is shown in Figure 13.

**Table 8.** Fuzzy set of input and output.

| $\Delta M_w$ | $e(\beta)$ | $e(\omega)$ |
|---|---|---|
| NB | NB | NB |
| NM | NS | NS |
| NS | ZE | ZE |
| ZE | PS | PS |
| PS | PB | PB |
| PM | | |
| PB | | |

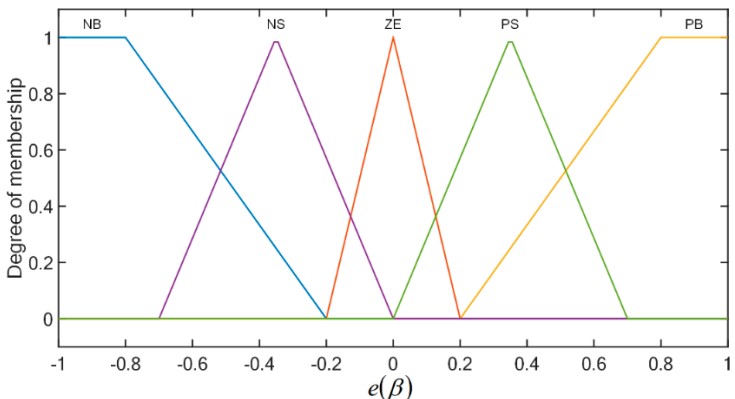

**Figure 11.** Membership function of $e(\beta)$.

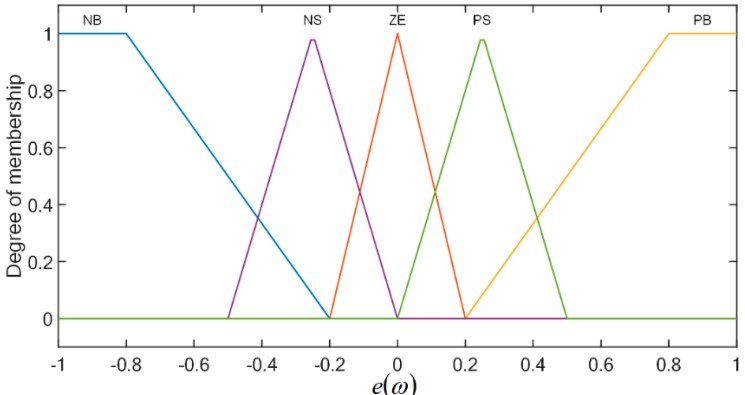

**Figure 12.** Membership function of $e(\omega)$.

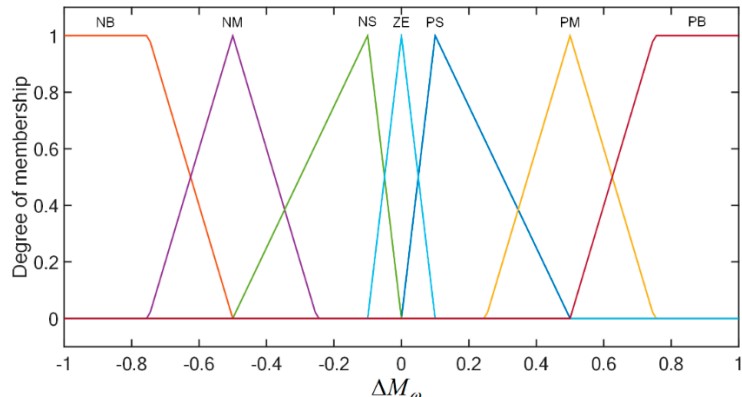

**Figure 13.** Membership function of $\Delta M_w$.

The fuzzy inference rule set is shown in Table 9.

**Table 9.** Fuzzy Inference Rule.

| $e(\omega)$ | $e(\beta)$ NB | NS | ZE | PS | PB |
|---|---|---|---|---|---|
| NB | NB | NB | NB | NM | NM |
| NS | NB | NM | NM | NS | NS |
| ZE | NS | NS | ZE | PS | PS |
| PS | PS | PS | PM | PM | PB |
| PB | PM | PM | PB | PB | PB |

*3.3. "PPS-Region" Based Dynamics Coordination Controller*

Based on the phase plane stability (PPS-region), the corresponding control strategies are developed for different states the vehicle is in, and the main control standards are shown in Figure 14.

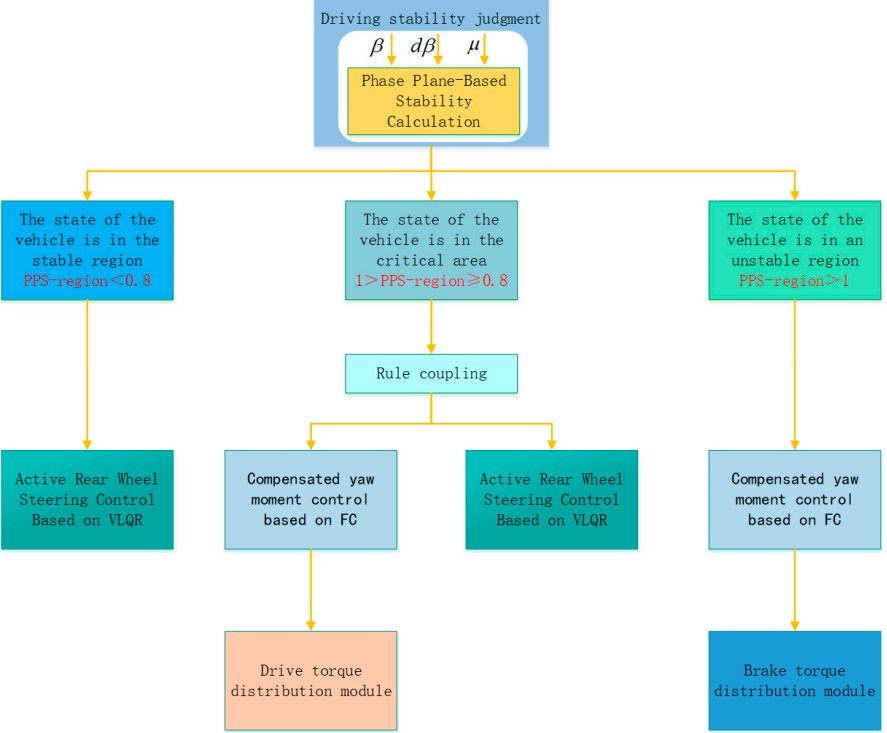

**Figure 14.** Coordinate the execution process of the control strategy.

In order to facilitate the execution of the coordinated control strategy, the active rear-wheel turning angle $\delta_r$ and the compensating yaw moment $\Delta M_w$ obtained from the previous calculation are assigned corresponding weighting coefficients to form the coordinated controller. The specific rules are shown in Table 10. The final calculation results of the coordinated controller are: vehicle active rear-wheel turning angle $\delta'_r$, drive compensating yaw moment $\Delta M'_d$, and brake compensating yaw moment $\Delta M'_b$.

**Table 10.** Coordinated control strategy.

| The Value Range of PPS-Region | $\delta'_r$ | $\Delta M'_d$ | $\Delta M'_b$ |
|---|---|---|---|
| (0, 0.8) | $\delta'_r$ | 0 | 0 |
| [0.8, 1] | $(1 - PPS - region)\delta_r$ | $PPS - region \cdot \Delta M_w$ | 0 |
| (1, $+\infty$) | 0 | 0 | $\Delta M_w$ |

3.3.1. Drive Force Distribution Module

The total longitudinal drive torque $\sum T_d$ of the vehicle is provided by the drive motors of all four wheels together. It is calculated by the formula

$$\sum T_d = k_{pd} T_0 \tag{20}$$

where: $k_{pd}$—Gas pedal opening; $T_0$—the maximum total drive force that the vehicle can provide, determined by the motor characteristics of the four-wheel hub motors. The formula for $T_0$ is

$$T_0 = \begin{cases} 1700 \, \text{N} \cdot \text{m}, \ n < 2000 \ \text{rpm} \\ \frac{2000}{n} * 1700 \, \text{N} \cdot \text{m}, \ n \geq 2000 \ \text{rpm} \end{cases} \tag{21}$$

where: $n$—rotational speed of the motor;

The following constraints should be satisfied between the total longitudinal driving moment $\sum T_d$, the compensating transverse moment $\Delta M_w$, and the driving force $T_{ij}$ of each wheel ($ij = fl, fr, rl, rr$) of the vehicle.

$$\begin{cases} \sum T_d = T_{fl} + T_{fr} + T_{rl} + T_{rr} \\ \Delta M_w = \dfrac{(T_{fl}+T_{fr})a \sin \delta_f + (T_{fr} \cos \delta_f - T_{fl} \cos \delta_f + T_{rr} - T_{rl})\frac{w}{2}}{R} \end{cases} \tag{22}$$

where: $w$—the distance between the two wheels of the rear axle; $R$—radius of the wheels. When the four wheels are in drive control, the steering angle of both rear wheels of the vehicle is 0. Therefore, the steering angle of the rear wheels is not reflected in Equation (22).

3.3.2. Braking Force Distribution Module

Under the actual driving conditions, if the braking torque is applied only for a single wheel, it will easily cause the controlled wheel to slip sideways. Therefore, to avoid this problem, the additional yaw moment required for each wheel should be allocated according to the load share of each wheel. The judgment and allocation rules are shown in Table 9.

Furthermore, the stability factor is one of the most significant parameters to characterize the stability performance of a vehicle and can characterize the steering state of the vehicle at a certain time. In this setting, it is easier for the driver to maneuver the vehicle when it is in neutral steering or appropriate understeering [36].

$$K = \frac{m}{L^2} \left( \frac{a}{C_r} - \frac{b}{C_f} \right) \tag{23}$$

where: $C_f$—cornering stiffness of the front axle; $C_r$—cornering stiffness of the rear axle; $L$-wheelbase, $L = a + b$;

When $K < 0$, the vehicle is in an oversteer state. When $K > 0$, the vehicle is in an understeer state.

In Table 11, $\omega_d$ is the expected value of the yaw rate, and $e_\omega$ is the difference between the expected and actual values of the side slip angle.

**Table 11.** Braking force distribution rules.

| $\omega_d$ | $e_\omega$ | Steering Status of the Vehicle | Wheels That Provide Braking Power |
|---|---|---|---|
| $\omega_d > 0$ Turn left | $e_\omega \geq 0$ | Oversteer | Right front wheel, Right rear wheel |
| | $e_\omega < 0$ | Understeer | Left front wheel, Left rear wheel |
| $\omega_d < 0$ Turn right | $e_\omega > 0$ | Understeer | Right front wheel, Right rear wheel |
| | $e_\omega \leq 0$ | Oversteer | Left front wheel, Left rear wheel |

When the left front wheel and the left rear wheel are used as braking wheels, the principle of braking torque distribution is

$$\Delta M_b' \approx \left( F_{fl} + F_{rl} \right) \times \frac{w}{2} \tag{24}$$

$$\begin{cases} T_{b\_fl} = \dfrac{F_{Z\_fl}}{F_{Z\_fl}+F_{Z\_rl}} \cdot \dfrac{\Delta M_b'}{\left(\frac{w}{2}\right)} \cdot R \\ T_{b\_rl} = \dfrac{F_{Z\_rl}}{F_{Z\_fl}+F_{Z\_rl}} \cdot \dfrac{\Delta M_b'}{\left(\frac{w}{2}\right)} \cdot R \end{cases} \tag{25}$$

When the right front wheel and the right rear wheel are used as the braking wheels, the principle of braking torque distribution is

$$\Delta M_b' \approx \left( F_{fr} + F_{rr} \right) \times \frac{w}{2} \tag{26}$$

$$\begin{cases} T_{b\_fr} = \dfrac{F_{Z\_fr}}{F_{Z\_fr}+F_{Z\_rr}} \cdot \dfrac{\Delta M_b'}{\left(\frac{w}{2}\right)} \cdot R \\[2mm] T_{b\_rr} = \dfrac{F_{Z\_rr}}{F_{Z\_fr}+F_{Z\_rr}} \cdot \dfrac{\Delta M_b'}{\left(\frac{w}{2}\right)} \cdot R \end{cases} \tag{27}$$

In Equations (24)–(27), $F_{ij}$ is the required braking force of each wheel; $T_{b\_ij}$ is the braking torque of each wheel; $\Delta M_b'$ is the braking compensating yaw moment; and $F_{Z\_ij}$ is the vertical load of each wheel ($i = ff, fr, rf, rr$, i.e., the abbreviations for left front, right front, left rear and right rear).

## 4. Hardware-in-the-Loop Simulation of Coordinated Control Strategies

### 4.1. Construction of Test Platform

The hardware-in-the-loop test platform built in this study mainly included the host computer, PXI chassis, DSP controller, and other vital components. The overall scheme of the aforementioned platform is shown in Figure 15.

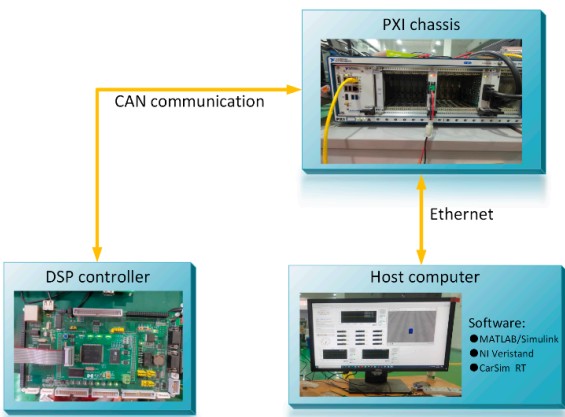

**Figure 15.** Hardware-in-the-loop test platform.

The host computer is mainly composed of PC-side computers equipped with software such as NI VeriStand, MATLAB/Simulink, and CarSim. During hardware-in-the-loop testing, download the CarSim vehicle dynamics model and some Simulink models to the PXI chassis. At the same time, the host computer acts to monitor various data in real-time and display the real-time driving animation of the vehicle provided by CarSim.

As a real-time running platform of various models, the PXI chassis is a transfer station for data interaction. It conducts real-time data exchange with the host computer through Ethernet communication. It conducts real-time data exchange with the DSP controller through the CAN communication network through the PXI-8512 board.

As the main body of the stability control program running in real-time, the DSP controller receives the accelerator/brake pedal signals sent from the vehicle VCU. At the same time, it will receive various vehicle driving parameters from CarSim sent by the PXI chassis to simulate the parameters collected by various sensors during the real vehicle driving process.

### 4.2. Hardware-in-the-Loop Simulation

In order to verify the feasibility and reliability of the coordinated control strategy, two test conditions, continuous gain sinusoidal response and double lane-change, are designed in this paper. The working conditions are shown in Table 12.

**Table 12.** Setting of test conditions.

| Types of Test Conditions | Initial Speed (km/h) | Input Charact Eristics of Front Wheel Steering Angle | The Road Adhesion Coefficient |
|---|---|---|---|
| Continuous gain sine test | 100 | Continuous sine gain with a period of 2 s and the maximum steering angle of 30° | 0.8 |
| Double lane-change test | 100 | After turning left back to positive, then right back to positive | 0.8 |

For the continuous gain sine test, a PID speed controller is used to maintain the vehicle speed at 100 km/h, and a continuous gain sine signal is used as the front wheel turning angle input. The test results of hardware-in-the-loop are shown in Figure 16.

The hardware-in-the-loop test platform built in this paper mainly includes the host computer, PXI chassis, DSP controller, and other vital components. The overall scheme of the hardware-in-the-loop test platform is shown in Figure 15.

It can be seen from Figure 16a that the vehicle under "ARS" control or coordinated control is able to dynamically calculate the active rear-wheel turning angle based on the vehicle's driving status. Before 4.5 s, the trajectories of the rear-wheel steering angles of the vehicles under "ARS" control and coordinated control are basically coincident because both vehicles are in the stable region and have the same control strategy. It can be seen from Figure 16b that the vehicle under coordinated control is capable of dynamically adjusting the wheel torque and using the yaw moment to control the vehicle. Figure 16c–f show that: The stability of the vehicle under "FWS" control is poor, with obvious speed drops at 3~4 s and 7~10 s, sharp fluctuations in the yaw rate, large tracking error in the side slip angle, and large displacement in the Y direction. The vehicle stability under coordinated control is satisfactory, the vehicle speed is basically maintained between 98 km/h and 102 km/h, the fluctuation of yaw rate is kept within $\pm 20°$/s, the tracking error of the side slip angle is consistently kept within $\pm 5°$, and the displacement in the Y direction is minor. The vehicle stability performance under "ARS" control is between the first two.

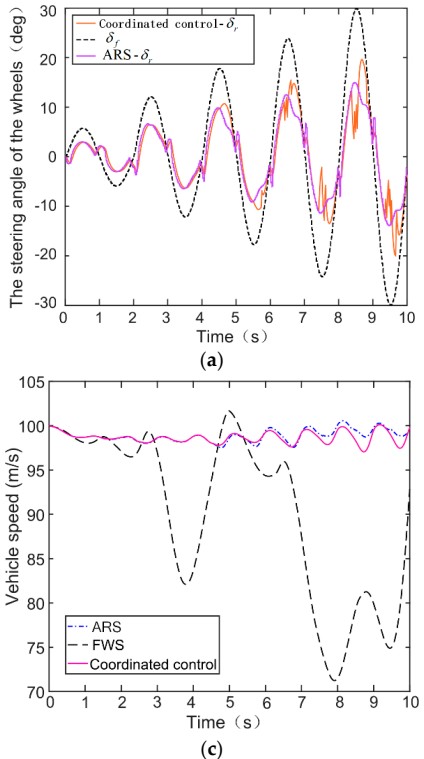
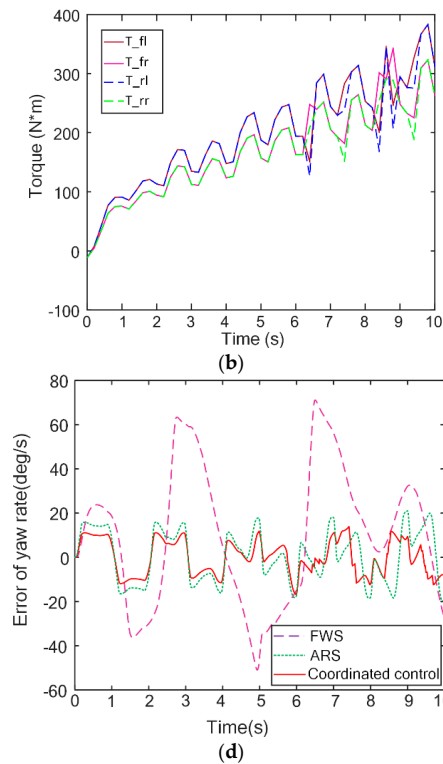

**Figure 16.** *Cont.*

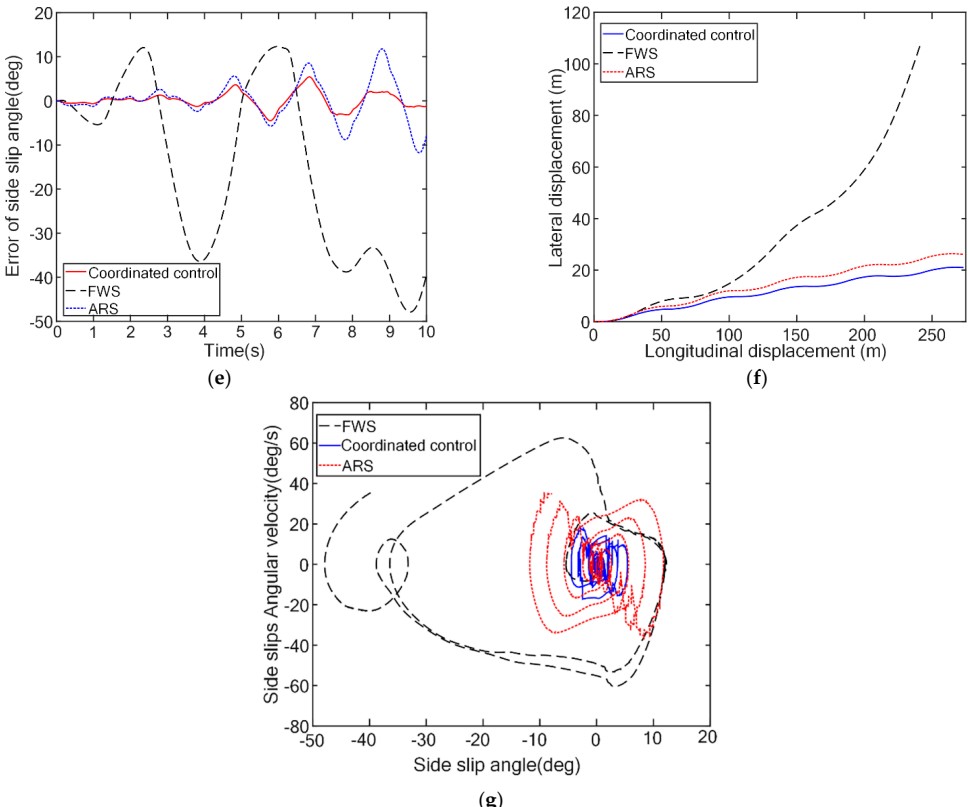

**Figure 16.** Hardware-in-the-loop test results under continuous sine gain conditions. (**a**) The curve of wheel steering angle change. (**b**) Wheel torque under coordinated control. (**c**) Variation curve of vehicle speed. (**d**) Response curve of the yaw rate. (**e**) Error response curve of the side slip angle. (**f**) Displacement of the center of mass of the vehicle. (**g**) Phase plane diagram of the side slip angle.

By analyzing the response curve of yaw rate error, it can be found that compared with the "FWS" control, under the control of "ARS" and "coordination strategy", the maximum amplitude of yaw rate error can be effectively reduced by 67% and 73%. The coordinated control can further reduce the maximum amplitude of the yaw rate error by 20% based on the control effect already achieved by the "ARS" control. By analyzing the response curve of the sideslip angle error, it can be found that compared with the "FWS" control, under the control of "ARS" and "coordination strategy", the maximum amplitude of the sideslip angle error can be effectively reduced by 60% and 85%. The coordinated control can further reduce the maximum amplitude of the side slip angle error by 62.5% based on the control effect already achieved by the "ARS" control.

Based on the above analysis, it is not difficult to find: Both "ARS" control and coordinated control can effectively reduce the maximum response amplitude of the yaw rate error curve and the maximum response amplitude of the side slip angle error curve. However, coordinated control can further optimize the control effect and better ensure the vehicle's stability under high-speed and complex working conditions. This conclusion can also be verified here from the phase plane diagram. It can be seen from Figure 16g that, compared with the other two control methods, the phase plane trajectory convergence of the vehicle under coordinated control is the finest.

As for the double lane-change test, the hardware-in-the-loop test results are shown in Figure 17.

The front-wheel steering angles of the three vehicles tracking the double shift trajectory and the active rear-wheel steering angles of the two vehicles under "ARS" control and coordinated control are shown in Figure 17a. Moreover, it can be seen from Figure 17b that under the test condition of double lane change, the vehicle under coordinated control can still dynamically adjust the wheel torque and use the yaw moment for control. Figure 17c–f

show that: The vehicle stability under "FWS" control is poor, with sharp fluctuations in the yaw rate and large tracking errors in the side slip angle. The vehicle stability under coordinated control is satisfactory. When the opening of the throttle pedal is kept constant, the vehicle speed is reduced to about 80 km/h, the yaw rate is consistently kept within ±15°/s, and the tracking error of the side slip angle is continuously kept within ±4°. The yaw rate error tracking performance of the vehicle under ARS control is in between the first two, and the effect of the side slip angle tracking error is most unfavorable. From Figure 17f, it can be seen that the driving trajectory of the vehicle under coordinated control fits best with the standard double lane-change trajectory ("DLC-path" in the figure). The vehicle is able to return to a stable driving condition as quickly as possible when it exits a double lane-change trajectory.

By analyzing the response curve of yaw rate error, it can be found that compared with "FWS" control, under the control of "ARS" and "coordination strategy", the maximum amplitude of yaw rate error can be effectively reduced by 55% and 68.6%. The coordinated control can further reduce the maximum amplitude of the yaw rate error by 40.6% on the basis of the control effect already achieved by the "ARS" control. By analyzing the response curve of the sideslip angle error, it can be found that compared with the "FWS" control, under the control of "ARS" and "coordination strategy", the maximum amplitude of the sideslip angle error can be effectively reduced by 23% and 57.4%. The coordinated control can further reduce the maximum amplitude of the side slip angle error by 44.7% on the basis of the control effect already achieved by the "ARS" control.

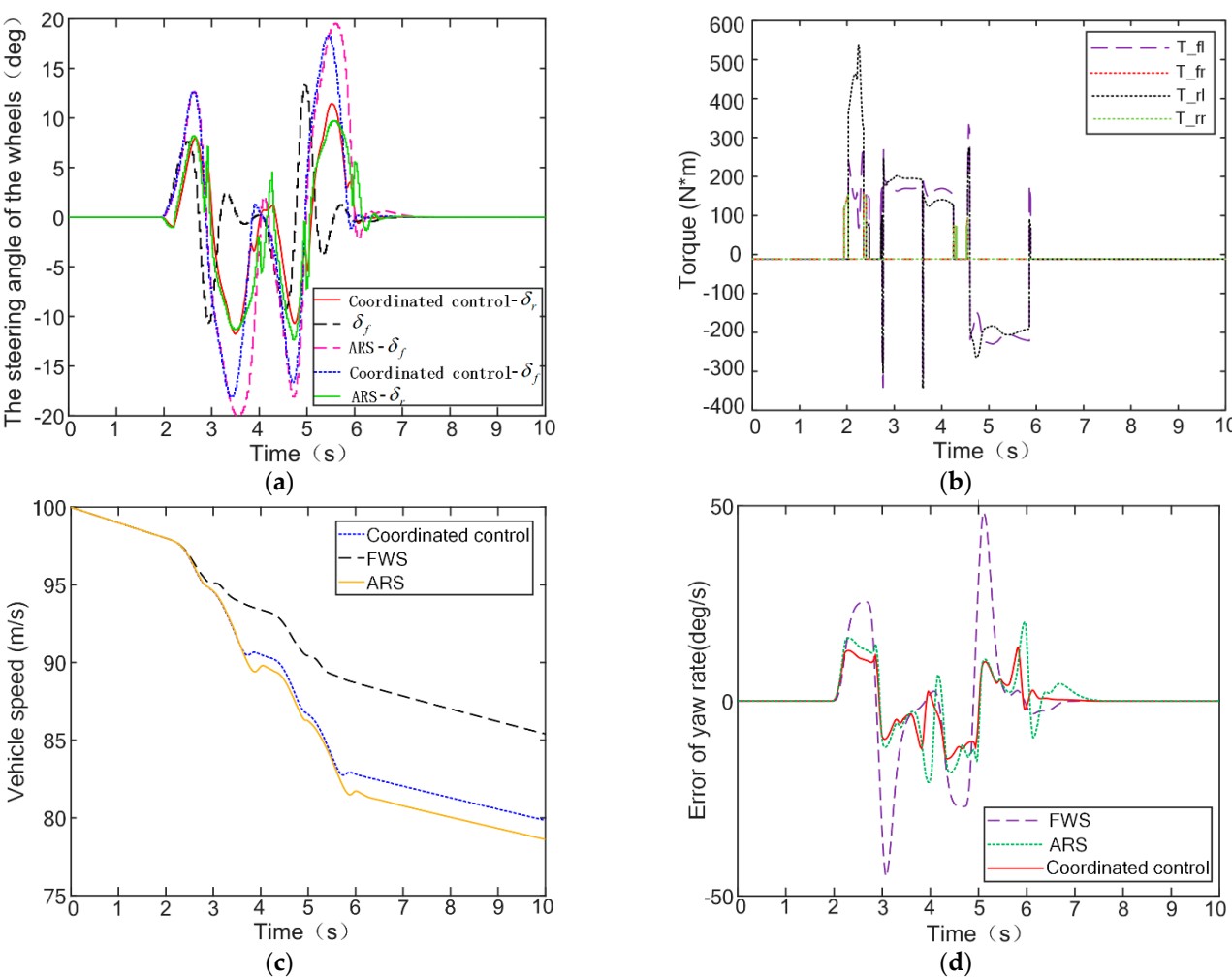

**Figure 17.** *Cont.*

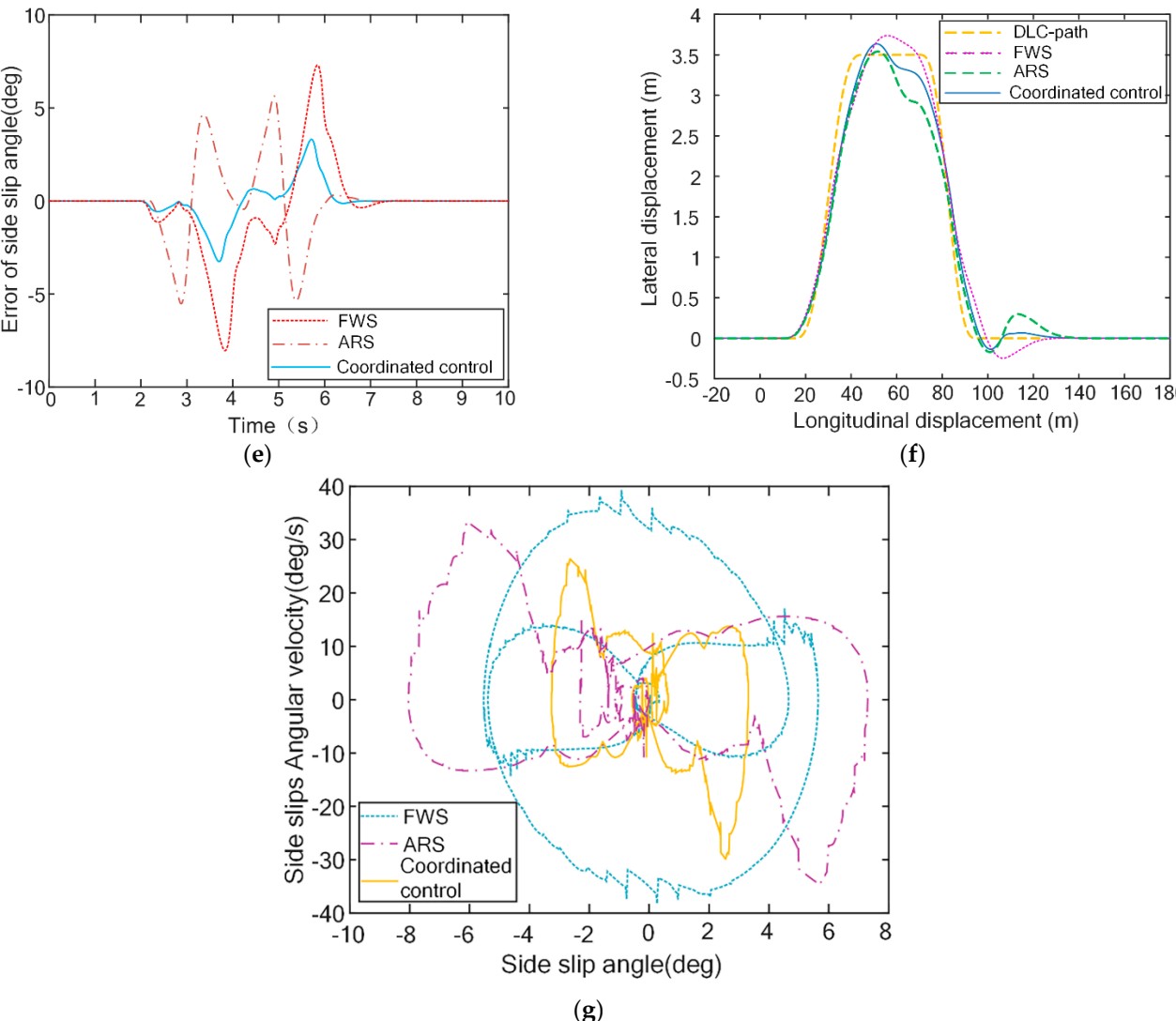

**Figure 17.** Hardware-in-the-loop test results under double lane-change test (**a**) The curve of wheel steering angle change. (**b**) Wheel torque under coordinated control. (**c**) Variation curve of vehicle speed. (**d**) Response curve of the yaw rate. (**e**) Error response curve of the side slip angle. (**f**) Displacement of the center of mass of the vehicle. (**g**) Phase plane diagram of the side slip angle.

Based on the above analysis, it is not difficult to find: Both "ARS" control and coordinated control can effectively reduce the maximum response amplitude of the yaw rate error curve and the maximum response amplitude of the side slip angle error curve. However, coordinated control can further optimize the control effect and better ensure the stability of the vehicle under high-speed and complex working conditions. This conclusion can also be verified here from the phase plane diagram. Figure 17g shows that, compared with the other two control methods, the phase plane trajectory convergence of the vehicle under coordinated control is the finest.

Combined with the above analysis of the tests, the coordinated control strategy greatly improves the vehicle's handling stability under double lane-change conditions and continuous sinusoidal gain conditions. The hardware-in-the-loop test results under two different experimental conditions demonstrate the rationality and effectiveness of the proposed coordination control strategy.

## 5. Conclusions

In this study, a dynamics coordinated control system was designed for a four-wheel-distributed driving and four-wheel-distributed steering electric vehicle. First, based on the phase plane division rule, the phase plane of "side slip angle-side slips angular velocity" was drawn, and the phase plane stability domain was divided according to the double-line method. Based on the stability domain boundaries, the further group plane was divided into a stable region, a critical region, and an unstable region. Furthermore, the phase plane stability index (PPS-region) quantitative characterization of vehicle stability conditions was introduced. This provided the basis for the design of the subsequent coordinated control. In this context, the coordinated control system included a VLQR-based active rear-wheel steering control strategy, an FC-based compensated yaw moment control strategy, and a PPS-region-based dynamics coordinated control strategy. The active rear-wheel steering controller was based on the LQR control controller, adding a fuzzy control variable weight coefficient regulator, through which the dynamic adjustment of the LQR controller's emphasis on the side-slip angle and the yaw rate was realized. Finally, the adaptability of the controller to changes in road conditions was enhanced. The compensated yaw moment controller was designed based on the fuzzy control theory. The coordination strategy was constructed according to the phase plane stability index PPS-region. Note that, according to the value of the PPS-region, the stability state of the vehicle was determined, and the corresponding control strategy was adopted.

Finally, to verify the effectiveness of the control strategy, a hardware-in-the-loop test platform was built, through which the hardware-in-the-loop test of the control algorithm was completed. The test results show that the control strategy correctly switches to the corresponding control strategy when the vehicle is in different stable states according to the divided steady domain state. Moreover, the phase plane curve of the vehicle responds quickly to return to the phase plane stable area as soon as possible, ensuring the handling performance and stability of the vehicle.

Under the test condition of continuous gain sine, compared with the "AFS" control, both the "ARS" control system and the coordinated control system can effectively reduce the yaw rate error curve amplitude of the vehicle at high speed. Under the action of the two control systems, the maximum amplitude of the yaw rate error curve is reduced by 67% and 73%, respectively. Among them, the coordinated control system can further improve the control effect of the "ARS" control system in terms of yaw rate by 20%. In terms of reducing the amplitude of the side slip angle error curve, compared with the "AFS" control, under the action of the "ARS" control system and the coordinated control system, the maximum amplitude of the error curve is reduced by 60% and 85%, respectively. It is worth noting that the coordinated control system can further improve the control effect of the "ARS" control system in terms of side slip angle by 20%.

Similarly, in the test condition of double lane-change, compared with the "AFS" control, the "ARS" control system and the coordinated control system can also effectively reduce the amplitude of the yaw rate error curve when the vehicle is traveling at high speed. Under the action of the two control systems, the maximum amplitude of the yaw rate error curve is reduced by 55% and 68.6%, respectively. Among them, the coordinated control system can further improve the control effect of the "ARS" control system in terms of yaw rate by 40.6%. In terms of reducing the amplitude of the side slip angle error curve, compared with the "AFS" control, under the action of the "ARS" control system and the coordinated control system, the maximum amplitude of the error curve is reduced by 23% and 57.4%, respectively. It is worth noting that the coordinated control system can further improve the control effect of the "ARS" control system in terms of side slip angle by 44.7%.

Further, in the above two test conditions, under the control of the coordinated control system, the phase plane curve of the vehicle can respond quickly and return to the phase plane stable area as soon as possible to ensure the handling performance and stability of the vehicle.

**Author Contributions:** Conceptualization, methodology, S.Z.; investigation, formal analysis, B.W. and D.L.; validation, writing—original draft preparation, B.W. and X.H.; supervision, writing—review and editing, H.C. and Y.Z.; funding acquisition, W.W. All authors have read and agreed to the published version of the manuscript.

**Funding:** We gratefully acknowledge the financial support of this research by the following projects: Control design of new energy vehicle air conditioning compressor based on intelligent multi-objective optimization (Grant No. ZDLQ2020002), Research on the theory and method of autonomous cooperative operation control of high-speed trains (Grant No. U1934221) and Research and manufacture of high precision grinding process of wear-resistant seals of construction machinery (Grant No. ZDLQ2021005).

**Data Availability Statement:** Not applicable.

**Conflicts of Interest:** The authors declare no conflict of interest.

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
