# Peer review of "A Dynamics Coordinated Control System for 4WD-4WS Electric Vehicles"

_electronics, doi:10.3390/electronics11223731_

Round 1
Reviewer 1 Report
In this manuscript, the authors proposed a A Dynamics Coordinated Control System for 4WD-4WS electric vehicles. The overall manuscript is well written.
• The present article provides the research community with an overview of dynamics coordinated control system based on fuzzy control theory for 4WD-4WS electric vehicles.
• Although the topic is important to the body of literature, but I have some concerns about the ‘scientific contributions’ of this manuscript. Authors are suggested to refine it to highlight the novelty of the work.
• Essentially, the technical discussion in this manuscript and the information is presented without a clear objective/foundation and motivation. Authors are suggested to add one motivation subsection in Introduction section.
• Authors are suggested to include comparative analysis table in section 1 for state-of-the-art approaches.
• A thorough proofreading/restructuring/grammar/sentence formation and spelling checking of this article is essential.
E.g. - "Then, to solve the coupling problem between the active rear-wheel steering controller and the compensating yaw moment controller in terms of control, based on the stability index PPS-region in the phase plane stability domain,a coordination rule is proposed" - in abstract, meaning is not clear and very long sentence.
• The article reports generic works and old references without any criticism or a clear message to conclude by the readers and add some latest references.
• In my opinion, it is more important to think of the applicability aspect of the technology and the associated enablers.
• The presented scheme is well presented but lack supportive references. In fact, one should expect a reasonable number of references in order to support the claims by literature. The main purpose of the article is to guide the research community and direct their attention to the most urgent/uncovered research areas in the desired field of research.
• In this manuscript, I highly recommend authors to include a detailed ‘comparative analysis of the proposed approach with existing approach’ that is thoroughly discussed and supported by reliable and up-to-date references.
• Authors need to justify that why they have employed fuzzy control theory not any other mechanism. Also, authors are suggested to include more results in section 4.
• The manuscript contains many abbreviations. Therefore, authors are suggested to include a table of abbreviations at the beginning of section 1 in order to improve the readability of the manuscript.
Reviewer 2 Report
The paper is interesting and useful for both engineers and researchers. It also has experimental validation of the proposed control strategy.
It is suggested to make comparison of the proposed control strategy with the existing solutions for four-wheel-distributed driving and four-wheel-distributed steering (4WD-13 4WS) electric vehicles.
The conclusions are somewhat similar to the abstract. The conclusions need to be better written. Rewrite the conclusions by clearly stating the advantages of the proposed control strategy.
Reviewer 3 Report
1. The contribution of this study has to be highlighted at the abstract section.
2. The contribution has to be stated clearly in bullets at the end of introduction section.
3. It is not understood why the authors have used Phase-Plane not Lypunove method to conduct the stability.
4. Most dynamic equations have not been cited as if they are developed by the authors.
5. The stability analysis has been assessed within linearized area and within local regions.
6. The authors have used Type-1 FL controller. I think it is instructive to use Type-2 Interval FL controller. Otherwise, this controller has to be indicated as future work. I suggest the work: DOI: 10.1177/0020294021997483
7. The results have to be supported by numerical evaluation or reported tables.
8. The description of the tools used in hardware-in-the-loop is weak.
9. The legends inside the figures have to be described within the text of manuscript.
10. The authors have to check the feasibility of generated torques in Figure (16).
11. The phase-plane plots have to be increased in resolution.
12. The comparison has been conducted with other control technques.
13. The conclusion is descriptive and it lacks to numerical assessment or improvement.
14. The used controller have to be compared to other controllers in the literature.
15. The number of references have to be updated.
Round 2
Reviewer 3 Report
The authors have addressed all my concerns. Thank you.